# Reliability and Accuracy of Linear Position Transducers During the Bench Press and Back Squat: Implications for Velocity-Based Training

**DOI:** 10.3390/jfmk10020109

**Published:** 2025-03-27

**Authors:** Raynier Montoro-Bombú, Armando Costa, Paulo Malico Sousa, Valter Pinheiro, Pedro Forte, Luis Monteiro, Alex S. Ribeiro, Luis Rama

**Affiliations:** 1Faculty of Sport Sciences and Physical Education, University of Coimbra, 3040-256 Coimbra, Portugal; 2Centro Interdisciplinar de Performance Humana (CIPER), Faculty of Human Kinetics, University of Lisbon, 1499-002 Cruz Quebrada, Portugal; 3Sports Sciences Laboratory (LAB-ISCE), Department of Sport Sciences, ISCE—Polytechnic University of Lisbon and Tagus Valley, 2620-379 Lisbon, Portugal; acosta_isce@hotmail.com (A.C.); direccaoddesporto@isce.pt (P.M.S.); prof_valterpinheiro@hotmail.com (V.P.); 4Live Quatity Research Center (LQRC), Complexo Andaluz, Apartado 279, 2001-904 Santarém, Portugal; 5Department of Sports, Higher Institute of Educational Sciences of the Douro, 4560-708 Penafiel, Portugal; pedromiguelforte@gmail.com; 6Department of Sports Sciences, Instituto Politécnico de Bragança, 5300-253 Bragança, Portugal; 7Research Center for Active Living and Wellbeing (Livewell), Instituto Politécnico de Bragança, 5300-253 Bragança, Portugal; 8Centro de Investigação em Desporto, Educação Física e Exercício e Saúde (CIDEFES), Lusófona University, 1749-024 Lisbon, Portugal; 9Faculty of Sport Science, Castilla-La Mancha University, 45071 Toledo, Spain; 10ICPOL, Higher Institute of Police Sciences and Internal Security, 1300-663 Lisbon, Portugal

**Keywords:** biomechanics, velocity-based training, force, monitoring training and testing, strength, performance

## Abstract

**Background:** Selecting the right linear position transducer (LPT) for velocity-based training monitoring sometimes presents uncertainties for coaches. **Objectives:** This study rigorously examined the test-retest reliability of three LPT–Cs using a simultaneous triangulation method of the same device during bench press (BP) and back squat (SQ) exercises performed on a Smith machine. **Methods:** Forty university students—13 females (23 ± 2 years) and 27 males (31.5 ± 6 years)—voluntarily participated in a randomized repeated-measures study. LPTs were randomly assigned numbers and placed at 5 cm apart to measure and collect bar displacement (∆S), mean propulsive velocity (MPV), peak velocity (PV), and time to peak velocity (T–PV). Each volunteer performed three BP and SQ attempts with pre-standardized loads (males: BP ≥ 40 kg and SQ ≥ 60 kg; females: BP ≥ 25 kg and SQ ≥ 40 kg). **Results:** The main findings of this study support a high degree of reliability for LPTs. For all variables, the absolute reliability presented significant values (*p* ≤ 0.05), with an intraclass correlation coefficient ≥ 0.995, a 95% confidence interval between 0.992–0.999, a coefficient of variation ≤ 10%, and a standard error of the mean ≤ 0.031. **Conclusions:** Scientists and coaches can use the LPT device as a reliable tool for monitoring velocity-based training by providing rigorous measurements of ∆S, MPV, PV, and T–PV during BP and SQ exercises. In addition, the smallest real difference reported may be useful in identifying minimal changes in ∆S within a single set (BP = 0.10 cm; SQ = 0.13 cm).

## 1. Introduction

Traditionally, adaptations to strength training have been monitored using a combination of different performance parameters, such as load intensity, exercise type and order, number of sets and repetitions (volume), rest between sets (density), and spatiotemporal control of movements (technique) [1]. Recent research showed that monitoring repetition velocity during strength training is also an objective and practical way to assess acute metabolic stress, hormonal response, and training-induced mechanical fatigue [1,2,3]. Research has focused more on the velocity of the concentric phase in an exercise, suggesting it as a valid approach for measuring and adjusting performance parameters [4]. This may be the fundamental reason why recent research has presented different devices that contribute to strength training control: accelerometers that measure acceleration [5,6], linear position transducers (LPTs) [6,7,8], and optical measurement systems (Trio–OptiTrack) [6].

For a measurement device to be recognized by scientists, trainers, and other users, it must show consistent results under the same conditions (repeatability) [6,9,10]. If this requirement is not met, the device may be useless in a practical context. The literature reveals several studies investigating validity and reliability conditions of different equipment [6,7,10,11,12]. For example, a systematic review analyzing multiple devices concluded that reliability analyses frequently used by researchers have not succeeded in differentiating between technological and biological variability, likely affecting the actual reliability of each device [12]. In most studies, the standard error of measurement (SEM) and the coefficient of variation (CV) were used to provide absolute inter-measurement reliability [1,6,7,10,11,12,13]. However, using the smallest real difference (SRD) has also been recommended, as it identifies whether an observed change score exceeds expected measurement error and represents a real change in measurement [10,14,15]. SRD reporting is considered a crucial issue in sports science studies, yet few have used it to analyze their results [10,11,16].

Another commonly used measure is the intraclass correlation coefficient (ICC) [1,10,11,13,17,18]. The ICC considers differences between measurements as discordance, whether they are constant or proportional. However, it was reported that the ICC alone can lead to errors in reliability analysis, making it necessary to declare confidence intervals (CI) as well [19]. Other studies have reported CCIs with 90% CIs, which can be confounding [20], while others do not report it, which sometimes brings more uncertainties [21]. Other published studies have used Pearson’s linear correlation coefficient to assess absolute reliability [11,21,22]. However, these were criticized because of the linear nature of the Pearson coefficient. This type of statistic may expose researchers to systematic errors in their interpretations, and more thorough analyses and rigorous interpretations may be necessary to ensure the reliability of the measurement devices [23,24]. Despite this high amount of available information, when the goal is to test the reproducibility of each device, standard reliability metrics typically include SEM, CV, and ICC.

Focusing on LPT reliability, we note that the validity and reliability of seven devices were recently analyzed using the Trio–OptiTrack as a gold standard [6]. The study concluded that the LPT called Speed4Lift, now known as Vitruve, demonstrated the best reliability. The authors also found that the T–Force LPT and the Chronojump LPT (LPT–C) presented lower reliability criteria than the Velowin and PowerLift LPTs [6]. A similar objective study found the T–Force LPT to be the most reliable technology for measuring barbell velocity during strength training and the only one recommended as a gold standard for comparing emerging technologies [10]. These results align with other reports that present positive reliability criteria for T–Force and Chronojump [7,13]. Specifically, the LPT–C employs a strap that connects directly to the barbell, allowing real-time measurement and collection of barbell displacement (∆S), mean propulsive velocity (MPV), peak velocity (PV), and time to peak velocity (T–PV), among others (Software—Chronojump 2.3.0-2423). These data, along with the athlete’s mass and/or the system mass, are used to calculate additional variables [11]. The LPT–C has been previously validated in comparison with T–Force [6,10,13] and Trio–OptiTrack [6]. Its reliability has also been analyzed [10,12,13,23] and widely used in different investigations [25,26,27,28]. However, uncertainty about the technological reliability of the LPT–C remains. No research has yet been conducted on how the reproducibility criteria also known as test-retest reliability and responsiveness behave when more than two devices are considered. All the studies referenced here have used one or two versions of the same device to analyze reliability. However, none of them have considered whether random error could have compromised the quality of the results. To eliminate uncertainties about the reliability of the LPT–C, the present study aimed to examine the test-retest reliability of three LPT–Cs using a simultaneous triangulation method of the same device in bench press (PB) and back squat (SQ) exercises performed on a Smith machine. Thus, this study aimed to assess the reliability of three LPT–Cs that simultaneously measured and collected ∆S, MPV, PV, and T–PV in BP and SQ exercises performed on a Smith machine using a recent linear position transducer. The use of the triangulation method with the same device has the potential to definitively determine the uncertainties regarding the reliability assumptions of this measurement tool. This approach could help obtain reliable values in control training and subsequent scientific research studies. It was hypothesized that the recent linear position transducer demonstrates reliability in PB and SQ exercises.

## 2. Materials and Methods

### 2.1. The Experimental Approach to the Problem

A randomized repeated measures study was conducted to examine the test-retest reliability of three LPT–Cs using a simultaneous triangulation method of the same device in bench press (BP) and back squat (SQ) exercises performed on a Smith machine. The LPT–Cs, spaced 5 cm apart, measured and collected ∆S and MPV, which was understood as the fraction of the concentric phase during which the measured acceleration was greater than the acceleration due to gravity [29]. PV was considered the highest instantaneous value of bar velocity during the concentric phase, and T–PV provided by the three devices. The LPT–C constituted the dependent variables, while BP and SQ exercises constituted the independent variables.

### 2.2. Subjects

Forty university students, 13 females (age: 23 ± 2 years, height: 171 ± 9.2 cm, weight: 74.9 ± 12.7 kg) and 27 males (age: 31.5 ± 6 years, height: 174 ± 0.09 cm, weight: 84.76 ± 9.67 kg) were recruited to participate in this study voluntarily. Inclusion criteria included: experience of more than one year of regular physical exercise and familiarity with the technical execution of BP and SQ. The 1RM was also a criterion (men: BP ≥ 40 kg and SQ ≥ 60 kg; women: BP ≥ 25 kg and SQ ≥ 40 kg). These criteria were checked before and during measurements. It was verified that no one had health problems, physical limitations, or musculoskeletal injuries that would compromise the performance of the measurements. Exclusion criteria applied if any of these requirements was unmet, resulting in six students being excluded, resulting in 40 subjects selected for reliability analysis. Subjects were instructed to avoid strenuous exercise and consumption of alcoholic beverages two days before the testing session. All participants were informed about the experimental procedures, as well as the potential harms and benefits of the study, and provided written consent. The research was conducted following the recommendations of the Helsinki Declaration (2013) and was approved by the scientific board and the ethics committee of ISCE—Polytechnic University of Lisbon and Tagus Valley, Department of Sport Sciences, 2620-379 Lisbon, Portugal (code-ISCE/EC/0022025 on 17 February 2024).

### 2.3. Measurement Equipment and Data Acquisition

A Smith machine (Multipower, Lisbon, Portugal) was used for all BP and SQ repetitions. This machine ensured that the eccentric and concentric displacements were only vertical, preventing possible measurement errors associated with horizontal bar displacement. The Smith machine had no counterweight mechanisms and functioned similarly to free weights (isoinertial loading). According to the manufacturer, the bar and the guide system weighed 22 kg. All weights placed at both ends of the bar were carefully calibrated using a SECA balance (Hamburg, Germany), which was also used to measure body mass. Height was measured with a stadiometer with an accuracy of 0.1 cm (Bodymeter 206, SECA, Hamburg, Germany). Body mass index was calculated according to previous protocols [30].

For data collection, the LPT–Cs were placed approximately 5 cm apart (Figure 1). The LPT–C (Chronojump, Barcelona, Spain) has a sensitivity of 1 mm at a frequency of 1000 Hz. The mean error for this device has been reported as 1.1 mm. The maximum cable length is 2.5 m, and the maximum detectable speed is 50 m/s (Chronojump 2.3.0-1507). The LPT–C was connected to a Chronopic 3.0 (Iearobotics) data acquisition system (Chronojump, Barcelona, Spain) and free Chronojump 2.3.0-2423 software provided by the company (Chronojump, Barcelona, Spain). The Chronojump software was installed on three 64-bit Windows computers. Each LPT–C was randomly numbered (Figure 1), allowing data collection and comparison across repetitions.

### 2.4. Testing Procedures

Data was collected at the Polytechnic University of Lisbon and Tagus Valley sports science laboratory. Before starting the measurements, anthropometric data were recorded. All subjects performed a 5-min warm-up on an exercise bike, followed by 5 min of mobility exercises for upper and lower limbs. They also performed two sets of five repetitions using only the barbell. Each student was instructed on the position to reach using a manual goniometer. This required continuous movement until the upper thighs were below the horizontal plane, with the knees flexed to a tibiofemoral angle of 35°–45° in the sagittal plane [10], before starting each data collection attempt. Each subject performed three attempts with pre-standardized loads (men: BP = 40 kg and SQ ≥ 60 kg; women: BP ≥ 25 kg and SQ ≥ 40 kg). Standardizing the weight aimed to obtain different loads for each subject, resulting in a greater tendency to variability in the data. Furthermore, by considering the individualization of the load based on relative intensity (% of 1RM), we could compromise the assumptions of data variability. This is because each percentage of 1RM has a range of associated mean propulsive velocity [8]. In the present study, the load was standardized to find different intensities and, therefore, testing the reliability of the devices at different velocities. The BP was performed in a supine position on a bench. During this exercise, the feet were placed on the floor, and the hands were placed on the bar. Subject positions and grip width were not standardized for both exercises, but a grip width of 5–7 cm away from the shoulders was recommended. The position on the bench was adjusted horizontally so that the vertical projection of the bar corresponded to the midline of each participant’s pectoral [10]. To standardize the executions and eliminate subjective timing aspects, during the eccentric phase, the bar was lowered in a controlled manner until it reached the nipples, where it remained isometric for approximately 1 s, eliminating the rebound of the bar on the chest. Then, the concentric phase was executed at the maximum achievable speed [10]. At the end of the concentric phase, lifting the shoulders and/or trunk from the bench was not allowed.

The SQ was performed in a traditional standing position with the bar placed on the trapezius, allowing external rotation of the feet (maximum 15°) parallel to the ground and separated at shoulder width (5–7 cm) [10]. During the eccentric phase, continuous movement was required until reaching a tibiofemoral angle of 35°–45°, at which point participants immediately transitioned to the concentric phase at the maximum achievable speed without taking their feet off the ground.

### 2.5. Statistical Analyses

For statistical analysis, only BP and SQ concentric actions were considered. All data followed a normal distribution and are reported as mean y standard deviation (mean ± SD). Using G*Power software version 3.1.9.7, statistical power was determined a priori based on differences between two dependent (pairwise) means. A beta value of 90% was reached, with an alpha of 0.05 and an effect size of 0.5, requiring a sample size of 36 participants. From the mean differences, a one-sample *t*-test was used to test whether the reported measurements between devices were statistically equal to zero. The same test was performed from the mean differences (MD) reported between devices. Acceptable absolute reliability was assessed by interactions for each paired LPT–C (i.e., LPT–C1 vs. LPT–C2 interaction, LPT–C1 vs. LPT–C3 interaction, and LPT–C2 vs. LPT–C3 interaction), following previous recommendations (CV >10% = poor, 5–10% = moderate, <5% = good) [31]. The standard error of the mean (SEM) was calculated from the MD [9]. The smallest real difference (SRD) between devices was calculated in Excel using the following equation: $1.96\times \sqrt{2}\times SEM$ [15]. The intraclass correlation coefficient (ICC) calculation was established following previous studies [32]. The ICC and 95% confidence intervals (CI) were based on the two-way mixed-effects model [23]. Thresholds for ICC were catagorized as poor (<0.5), moderate (0.5–0.75), good (0.75–0.9), and excellent (<0.9) [23]. The alpha was set at *p* ≤ 0.05. Potential biases were analyzed using a simple linear regression model (SLR). For this analysis, we considered the mean and MD data. Data analysis was performed using the statistical program SPSS, V.28.0, and graphs were produced using GraphPad Prism software, Version 9.4.0.

## 3. Results

The mean values (mean ± SD) of reproducibility for each analyzed variable are shown in Table 1. Based on the T-test data, we found that most of the measurements reported between devices were statistically equal to zero (*p* ≤ 0.05), representing the first step in assessing inter-device reliability. We found only during the MPV of BP that the LPT–C1 vs. LPT–C2 interaction was statistically different from zero (*p* = 0.116) (Table 2).

When analyzing the LPT–C1 vs. LPT–C3 interaction in BP, we identified that MPV presented values that partially compromised absolute reliability (CV = 10.16). However, this was not observed in the remaining interactions, which presented excellent degrees of absolute reliability (LPT–C1 vs. LPT–C2, CV = 3.10%; LPT–C2 vs. LPT–C3, CV = 3.73%). The same T–PV interaction presented moderate absolute reliability (CV = 8.11), contrary to the rest of the interactions that showed excellent reliability (CV ≤ 4.61). The rest of the interactions showed excellent absolute reliability (Table 2). Additionally, a relatively small proportional bias for ∆S (*p* = 0.009) was also present during the LPT–C1 vs. LPT–C2 interaction in BP, though this bias was not identified in the rest of the interactions measuring ∆S in the same exercise (LPT–C1 vs. LPT–C3, *p* = 0.500; LPT–C2 vs. LPT–C3, *p* = 0.848). Despite these observed biases, according to the criteria for the remaining interactions and the SEM results (Figure 2, Figure 3, Figure 4 and Figure 5), the overall performance of LPT–C shows robust absolute reliability. These test-retest reliability assumptions increase when analyzing the near-perfect intraclass correlation thresholds (ICC ≥ 0.995; CI between 0.992–0.999) for the entire set of analyzed metrics (Figure 6, Figure 7, Figure 8 and Figure 9).

## 4. Discussion

This study examined the test-retest reliability of three LPT–Cs that simultaneously measured and collected ∆S, MPV, PV, and T–PV in BP and SQ exercises performed on a Smith machine. Using this triangulation, we aimed to ensure that reliability assumptions were not the product of only two devices, making absolute agreement analyses more robust. Our findings provide a strong statement on the reliability of this tool, given that the SEM and CV values found for each variable are considered. To our knowledge, this is the first study to analyze test-retest reliability using three LPT–Cs as references. The main findings indicate a high degree of reliability for this device, with excellent ICC, SEM, and CV values. The ICCs are almost perfect, with the lowest identified in the LPT–C1 vs. LPT–C2 interaction during the MVP in SQ (ICC= 0.995; CI = 0.990–0.997) and in the LPT–C2 vs. LPT–C3 interaction (ICC= 0.995; CI = 0.995–0.997). A relatively small proportional bias was also found for ∆S (*p* = 0.009) during the LPT–C1 vs. LPT–C2 interaction in BP, though this bias was not identified in the rest of the ∆S interactions for the same exercise (LPT–C1 vs. LPT–C3, *p* = 0.500; LPT–C2 vs. LPT–C3, *p* = 0.848). Therefore, our results support the idea that LPT–C is reliable for monitoring speed-based training, confirming the hypothesis of this study.

When analyzing the PV and MPV in both exercises, we found that the MD among the LPT–Cs was considerably lower than in the remaining metrics. The CIs found were also substantially low and did not show criteria representing a large dispersion of the data (Table 2). The CVs for BP and SQ showed absolute reliability for most of the interactions analyzed. However, absolute reliability results for BP MPV between LPT–C1 vs. LPT–C3 appeared poor (CV = 10.16) [31]. To interpret these assumptions within the same interaction and ensure the result was not due to data collection errors, we first confirmed that reported measurements between devices were statistically equal to zero *p* = <0.001, a fundamental requirement for reliability analysis. We then verified that the SEM was proportionally lower than previous reports [7,10]. Another criterion supporting our results is the SRD. Previous studies hold that the SEM should be considered the minimum threshold for using a specific device [9,17]. However, the SDC is ideal for identifying significant changes in performance and assessing actual effort during training [7]. In our study, this variable presents values that can be considered very low (Table 2). We did not find proportion biases in any of the exercises for PV. The SDC and CV we found for SQ MPV and PV show smaller mean values than previously reported [7,10].

We reported high absolute reliability (ICC ± CI) between LPT–Cs [23]. Each metric independently showed ICC values ≥ 0.995, with CIs between 0.990–0.999 (Figure 5, Figure 6, Figure 7, Figure 8). These results could also be expected due to the high sample variability, a fundamental requirement for this type of study [23]. When independently analyzing the reliability plots, we found that the values constitute a near-perfect linear distribution. This indicates the high reproducibility of the values regardless of distance, speed, and high or low times. These results align with previous studies that compared the same equipment but with only two encoders as a reference. When comparing the BP, researchers reported similar results for MPV (ICC = 0.995; CI = 0.992–0.995) and PV (ICC = 1.000; CI = 0.999–1.000). These results were also analogous when comparing the SQ [10]. Studies with other devices (only two T–force) found perfect intra-device reliability during PB and SQ, where PV (ICC = 1.000, CI = 0.999–1.000; SEM = 0.01; CV% = 0.45), with similar values also found for MPV. In the same study, the results found for the Vitruve device also presented recurrent data indicating high reliability [7]. As far as we were able to review, we have not seen other studies presenting comparable ICC data for our ∆S and T–PV results. Only one study analyzed ∆S, but it reported standard errors as a percentage of the 90% CI, so these results are not comparable [11].

The ∆S is an essential measure of performance parameters [33]. It is common that with modification of displacement (which causes a change in range of motion), there can be modifications in force-length and force-velocity relationships [34], as well as power output [35]. Several studies have compared the mean barbell velocity in different exercises. However, none of those cited here have reported the value of mean barbell velocity under the displacement conditions. This criterion may be necessary because two repetitions with equal load (80 kg) and equal average velocity (0.98 m/s) but with different ∆S can significantly affect both the quality and intensity of execution.

Furthermore, increased ∆S is associated with greater strength and hypertrophy gains [36]. Considering these criteria, ∆S is a metric that coaches and performance physiologists should constantly analyze. Devices showing high degrees of validity and reliability may be necessary to measure ∆S. Overall, our results indicate high reliability among LPT–Cs. The CVs are good for both PB and SQ, and the ICC values are slightly lower in PB than in SQ (Table 2). No systematic or proportional biases were found in the iterations (Table 2). The MD during ∆S was higher than the remaining metrics, and between LPT–C1 vs. LPT–C2, the highest values were found for both PB and SQ (Table 2). For PB MD (0.31 ± 0.47; CI 0.24–0.37), this barely represents, in practical terms, a difference of 3 mm, which is in complete correspondence with the results reported in the Figure 1. We recommend that practitioners use this SEM data as the range within which a displacement error (0.3 mm) might be expected. In practical training terms, this may not be representative for coaches, but it may be significant for a sport scientist considering this measure as a significant difference for their research. We also recommend using the SDC criteria reported here to identify the minimum change in ∆S within a series (BP = 0.10 cm; SQ = 0.13 cm).

Numerous sports require high-force production in a very short time [37]. In this sense, T–PV can be a key indicator for these disciplines [38]. The T–PV varies depending on the type of exercise, the load used, and the athlete’s training level. Based on this criterion, it is consistent to think that decreasing T–PV without affecting the magnitude of peak force results in an increase in velocity and, consequently, mechanical power and the rate of force development, which is typically measured at any instant on the slope of the force-time curve (Δforce/Δtime) [39]. Previously, it was documented that T–PV can be responsible for expressing high power output in different exercises [39] and is frequently used in biomechanical studies [40]. Technological devices must be used to measure T–PV. Therefore, the correct quantification and interpretation of T–PV depend on the reliability of the equipment used and are crucial for researchers, training, and sports rehabilitation professionals. In our study, T–PV also showed high reliability (ICC ± 95% CI) (see Figure 9). The MD found here was considerably lower than ∆S and MPV but higher than PV overall. The moderate to good CVs (Table 2) support the high reliability of LPT–C for measuring T–PV. We also found no tendency to underestimate and/or overestimate the data for this variable. As far as we could review, we found no studies in the available literature with contrasting results to those found here.

Regardless of the results, some limitations and orientations for future research should be considered. Although we only analyzed MPVs in this study, it should be noted that other velocities can be obtained with LPT–C. The mean velocity, due to the values between the start and end of the movement, can also be considered for reliability analysis in future research. Although other studies have shown its reliability [7,10], analyzing its behavior with three devices would be more rigorous. It is important to highlight that MPV was the variable used since, for more than 80% of the participants, the standardized weight (men: BP ≥ 40 kg and SQ ≥ 60 kg; women: BP ≥ 30 kg and SQ ≥ 40 kg) represented between 40% and 70% of their RM. Therefore, we had to follow previous recommendations [29]. As expected, our data responded to variability patterns, likely because the recruited subjects were from a university population. Still, we strongly recommend against using this type of design with highly trained athletes, where data variability may be compromised. Finally, we found underlying mobility difficulties among the participants, mainly in the SQ exercise. This certainly does not represent a measurement error, as data collection was simultaneous, but it does represent a protocol limitation. Future research can use this device with a high level of reliability as long as the criteria discussed here are also considered.

## 5. Conclusions

This study rigorously examined test-retest reliability through the triangulation of the same device. The main findings of this study support a high degree of reliability of the LPT–C, with ICC, CV, and SEM results as references. This suggests that the LPT–Ccan be introduced as a reliable tool for measuring ∆S, MPV, PV, and T–PV in BP and SQ exercises. It can also be used during velocity-based strength training without incurring reliability errors between low and high loads. The SDC criteria reported here can be helpful for coaches and sports scientists who want to identify the minimum changes in ∆S within a set, i.e., when one repetition can be considered different from the other (BP = 0.10 cm; SQ = 0.13 cm).

## Figures and Tables

**Figure 1 jfmk-10-00109-f001:**
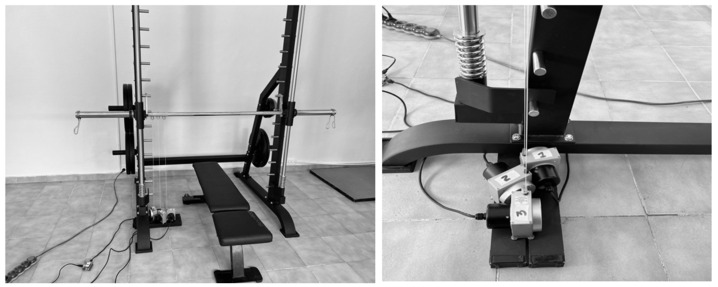
Placement of data collection devices.

**Figure 2 jfmk-10-00109-f002:**
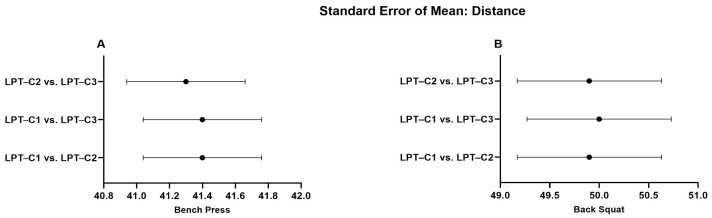
Standard error of distance measurement. (**A**) Bench Press; (**B**) Back Squat. LPT–C = Chronojump linear position transducer.

**Figure 3 jfmk-10-00109-f003:**
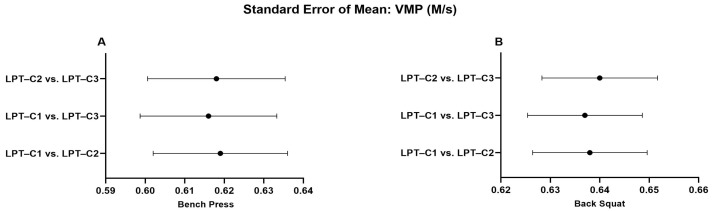
Standard error of measurement for mean propulsive velocity (MPV). (**A**) Bench Press; (**B**) Back Squat. LPT–C = Chronojump linear position transducer.

**Figure 4 jfmk-10-00109-f004:**
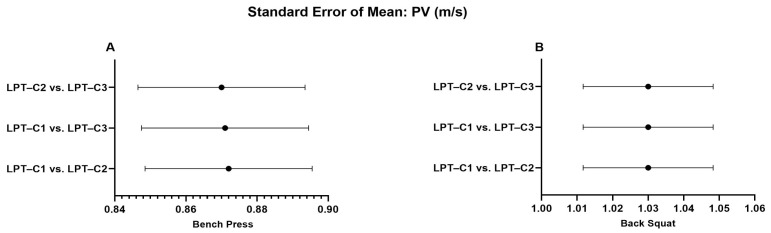
Standard error of measurement for mean peak velocity (PV). (**A**) Bench Press; (**B**) Back Squat. LPT–C = Chronojump linear position transducer.

**Figure 5 jfmk-10-00109-f005:**
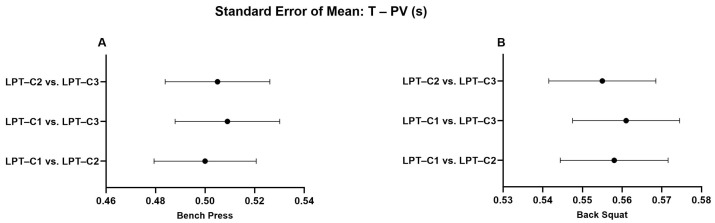
Standard error of measurement for time to peak velocity (T–PV). (**A**) Bench Press; (**B**) Back Squat. LPT–C= Chronojump linear position transducer.

**Figure 6 jfmk-10-00109-f006:**
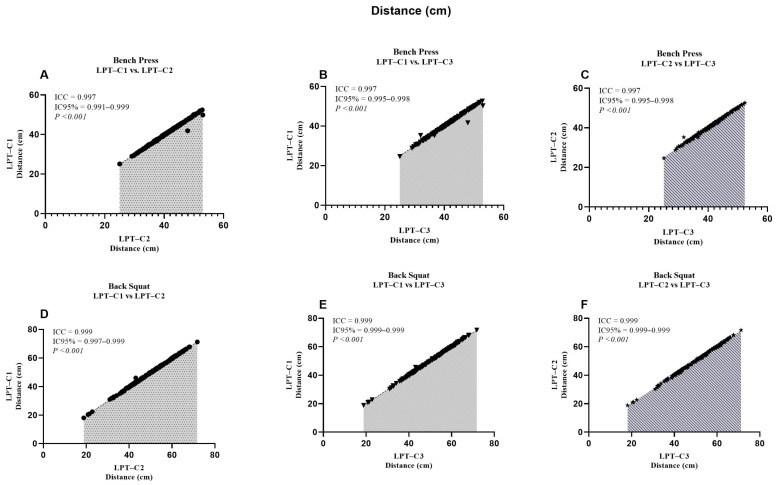
Reliability values of the distance between the adjusted interactions for both exercises. ICC = Intraclass correlation coefficient; IC95% = confidence interval of 95%; LPT–C= linear position transducer of Chronojump.

**Figure 7 jfmk-10-00109-f007:**
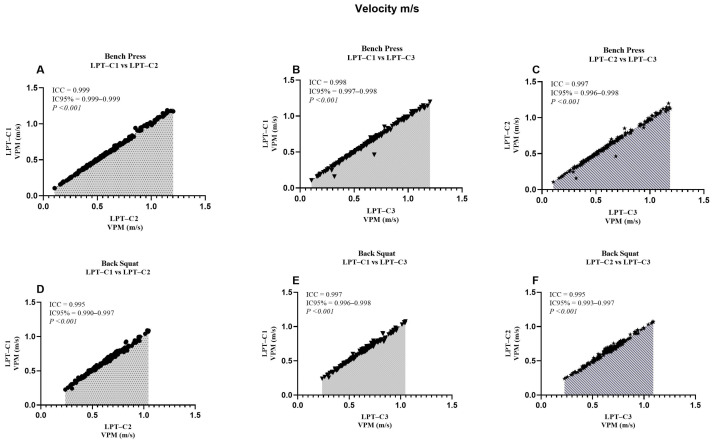
Reliability values of the mean propulsive velocity between the adjusted interactions for both exercises. ICC = Intraclass correlation coefficient; CI95% = confidence interval of 95%; LPT–C= linear position transducer of Chronojump.

**Figure 8 jfmk-10-00109-f008:**
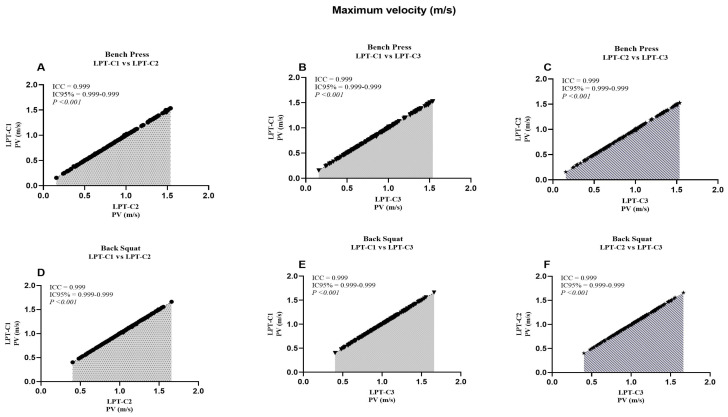
Reliability values of the peak velocity between the adjusted interactions for both exercises. ICC = Intraclass correlation coefficient; CI95% = confidence interval of 95%; LPT–C= linear position transducer of Chronojump.

**Figure 9 jfmk-10-00109-f009:**
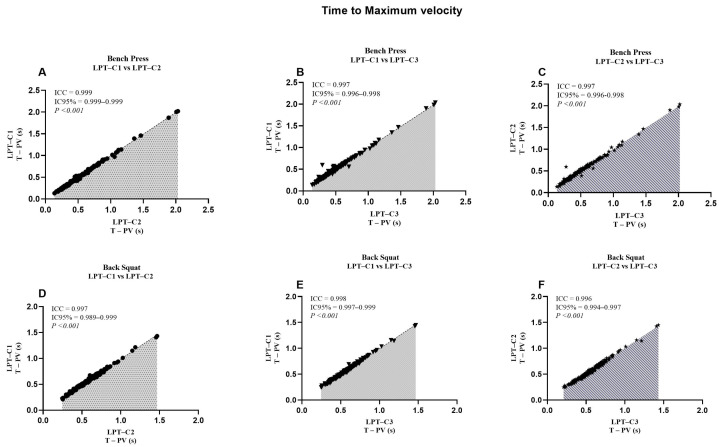
Reliability values of the time to peak velocity between the adjusted interactions for both exercises. ICC = Intraclass correlation coefficient; CI95% = confidence interval of 95%; LPT–C= linear position transducer of Chronojump.

**Table 1 jfmk-10-00109-t001:** Data (mean ± SD) of intra-device reproducibility.

	Bench Press (BP)			Back Squat (SQ)		
	Encoder–1	Encoder–2	Encoder–3	Encoder–1	Encoder–2	Encoder–3
∆S (cm)	41.5 ± 5.20	41.2 ± 5.11	41.4 ± 5.17	50.1 ± 10.4	49.8 ± 10.4	50.0 ± 10.4
MPV (m/s)	0.617 ± 0.245	0.621 ± 0.249	0.614 ± 0.245	0.634 ± 0.163	0.621 ± 0.173	0.614 ± 0.165
PV (m/s)	0.872 ± 0.333	0.871 ± 0.332	0.869 ± 0.331	1.03 ± 0.259	1.03 ± 0.260	1.03 ± 0.259
T–PV (s)	0.504 ± 0.291	0.496 ± 0.291	0.514 ± 0.326	0.563 ± 0.192	0.551 ± 0.192	0.559 ± 0.191

∆S = bar displacement; MPV = mean propulsive velocity; PV = peak velocity; T–PV = time to peak velocity.

**Table 2 jfmk-10-00109-t002:** Absolute concordance values between LPT–Cof the same brand during the bench press and back squat exercises.

		Bench Press (BP)			Back Squat (SQ)		
		LPT–C1 vs. LPT–C2	LPT–C1 vs. LPT–C3	LPT–C2 vs. LPT–C3	LPT–C1 vs. LPT–C2	LPT–C1 vs. LPT–C3	LPT–C2 vs. LPT–C3
**∆S (cm)**	MD	0.31 ± 0.47	0.18 ± 0.56	0.12 ± 0.37	0.31 ± 0.32	0.10 ± 0.30	0.20 ± 0.43
	CI-95% upper	0.37	0.26	0.18	0.35	0.148	0.145
	CI-95% lower	0.24	0.10	0.07	0.26	0.064	0.026
	T-Test (*p*)	<0.001	<0.001	<0.001	<0.001	<0.001	<0.001
	SLR	0.009	0.500	0.848	0.391	0.549	0.540
	CV(%)	1.52	3.05	2.96	1.05	2.82	2.09
	SRD (m s^−1^)	0.093	0.110	0.110	0.124	0.158	0.148
**MPV (m/s)**	MD	0.004 ± 0.014	0.002 ± 0.023	0.006 ± 0.025	0.010 ± 0.019	0.004 ± 0.018	0.006 ± 0.022
	CI-95% upper	0.006	0.005	0.0010	0.013	0.006	0.009
	CI-95% lower	0.002	0.000	0.003	0.008	0.001	0.003
	T-Test (*p*)	<0.001	<0.001	0.116	<0.001	<0.001	<0.001
	SLR	0.534	0.990	0.963	0.401	0.193	0.102
	CV(%)	3.10	10.16	3.73	0.26	0.27	0.26
	SRD (m s^−1^)	0.004	0.004	0.004	0.013	0.003	0.013
**PV (m/s)**	MD	0.001 ± 0.005	0.003 ± 0.005	0.001 ± 0.006	0.001 ± 0.003	0.001 ± 0.004	0.001 ± 0.004
	CI-95% Upper	0.002	0.003	0.003	0.001	0.006	0.002
	CI-95% lower	0.000	0.002	0.000	0.000	0.002	0.001
	T-Test (*p*)	<0.001	<0.001	<0.001	0.056	<0.001	<0.001
	SLR	0.313	0.456	0.524	0.152	0.168	0.346
	CV(%)	3.13	1.81	4.20	3	2.99	4.00
	SRD (m s^−1^)	0.001	0.001	0.001	0.000	0.000	0.001
**T–PV (s)**	MD	0.007 ± 0.016	0.002 ± 0.021	0.005 ± 0.023	0.012 ± 0.017	0.004 ± 0.016	0.007 ± 0.022
	CI-95% upper	0.010	0.005	0.008	0.015	0.006	0.011
	CI-95% lower	0.005	0.000	0.001	0.010	0.002	0.004
	T-Test (*p*)	<0.001	0.008	0.003	<0.001	<0.001	<0.001
	SLR	0.575	0.433	0.256	0.570	0.593	0.334
	CV (%)	2.15	8.11	4.61	1.36	3.48	2.92
	SRD (m s^−1^)	0.003	0.032	0.032	0.003	0.003	0.004

LPT–C= Chronojump position linear transducer; ∆S = bar displacement; MPV = mean propulsive velocity; PV = peak velocity; T–PV = time to peak velocity; MD = mean differences; SRD = smallest real difference; CV = coefficient variation; CI-95% = 95% confidence intervals; SLR = simple linear regression model.

## Data Availability

The data supporting this study’s findings are available from the first and corresponding author upon reasonable request.

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
