# Peer review of "Reliability and Accuracy of Linear Position Transducers During the Bench Press and Back Squat: Implications for Velocity-Based Training"

_jfmk, 2025, doi:10.3390/jfmk10020109_

Round 1
Reviewer 1 Report
Comments and Suggestions for Authors
I suggest Minor revisions.
See pdf. file
Kind regards.

Author Response
The authors are grateful for the comments made by the reviewer. The recommendations made were accepted and included in the body of the manuscript.
Line 48 reference.
Response/ Thank you very much for your comment, this was included in the official manuscript body.
Line 126-127 Present height data in centimeters
Response/ Thank you very much for your comment. We have corrected it in the body of the manuscript.
Line 303 You said previous studies. Please list them.
Response/ Thank you very much for your comment. We have corrected it in the body of the manuscript.
Line 401 Put a point after journal abbreviations. for example J. Strength Cond. Res.
Thank you very much for your comment. We have corrected it in the body of the manuscript.
Line 408 abbreviations Journal names should be written as abbreviations
Response/ I appreciate your comment. We have corrected it in the body of the manuscript.
Line 411 abbreviations
Response/ I am grateful for your comment. We have corrected it in the body of the manuscript.
Line 418 What is doi number?
Response/ I am grateful for your comment. We have corrected it in the body of the manuscript.
Reviewer 2 Report
Comments and Suggestions for Authors
Thank you for the opportunity to review this manuscript, which considers some interesting, applied issues.
This study appears to be novel, and author showed an interesting point about “Reliability, Accuracy and
Smallest Detectable Change of Linear Position Transducers During the Bench Press and Back Squat:
Implications for Monitoring Training and Testing”.
The manuscript is written in accordance with the journal's guidelines, contains an adequate paper
structure, and all chapters are very concise and easy to read.
The study addresses an important topic in sports science by evaluating the reliability of linear position
transducers (LPT) for velocity-based training. The results provide valuable insights for coaches and
researchers, confirming LPT devices as accurate tools for monitoring strength training.
The study’s findings are highly relevant for practical applications, helping improve performance
assessment in resistance training.
Based on what I have read, I notice a few things that would be good to correct, in order to improve the
quality of the article. Please see my comments below.
Replace the word "monitoring" from the keywords, as it is already in the manuscript title.
Figures are blurry and of low quality when zoomed in. I ask authors to replace figures in accordance with
the journal's figure quality guidelines.
The sample is limited to university students, which may affect the generalizability of the findings to elite
athletes. I suggest that the authors acknowledge this limitation in the manuscript and propose future
studies that include highly trained individuals to determine whether the reliability of LPTs remains
consistent across different performance levels.
I urge the authors to conduct a more comprehensive review of the recent literature and integrate studies
published within the last five years. This will not only strengthen the theoretical framework and discussion
but also enhance the manuscript’s overall impact and reliability.
Author Response
Thank you very much for your interesting comments and contributions to improving the manuscript. The authors appreciate the value of the comments and recognize that they were important in consolidating the paper. We will also try to satisfy each of the questions.
Replace the word "monitoring" from the keywords, as it is already in the manuscript title.
Response/ Thank you for your comment, this has been removed.
Figures are blurry and of low quality when zoomed in. I ask authors to replace figures in accordance with the journal's figure quality guidelines.
Response/ Thank you very much for your comment. The images were extracted directly from the Prisma GraphPad program at a resolution of 300 dpi and a 24-bit color model. In Image 1, we have included a detailed explanation of each image's purpose. To improve the quality of the images, could you specify which image you are referring to? This would help us identify the issue and provide a prompt solution.
The sample is limited to university students, which may affect the generalizability of the findings to elite athletes. I suggest that the authors acknowledge this limitation in the manuscript and propose future studies that include highly trained individuals to determine whether the reliability of LPTs remains consistent across different performance levels.
Response/ We thank the reviewer for their comments. We acknowledge that generalising results across different populations can be a fundamental challenge in various investigations. However, we do not believe this issue directly affects the results of our study. In our case, the study design allowed for high data variability—an essential requirement for this type of research. As a result, all three devices analyzed demonstrated a high degree of reliability at both high and low speeds. Therefore, we believe our findings can be generalised to any population, regardless of whether they are athletes or non-athletes.
I urge the authors to conduct a more comprehensive review of the recent literature and integrate studies published within the last five years. This will not only strengthen the theoretical framework and discusión but also enhance the manuscript’s overall impact and reliability.
Response/ We would like to express our gratitude to the reviewer for this comment. A comprehensive review of the entire manuscript was conducted, leading to the identification of an optimal percentage distribution of the references used. Specifically, out of the 40 total references, 35% (14 references) are from the years 2025 to 2021, while another 35% (14 references) come from the recent decade (2020 to 2015). Only 10% (4 references) date from 2015 to 2010, and 20% (8 references) cite classical authors whose scientific contributions are widely recognized. However, we are unclear on how the inclusion of additional references would enhance the reliability and impact of our results
Reviewer 3 Report
Comments and Suggestions for Authors
First of all, thank you for giving me the opportunity to review this article and provide my critical perspective on it. I would like to congratulate the authors on this study, which investigates the reliability of linear position transducers for measuring key parameters of velocity-based training in exercises such as the bench press and squat. This article is well-structured, has a solid justification, and contributes valuable data to the field of strength training. A major strength is the use of three linear position transducers simultaneously to triangulate the measurements. However, the following aspects for improvement should be considered:
Title
The title is informative, but I find it too long. I recommend that the authors summarize it, for example:
"Reliability and Accuracy of Linear Position Transducers in Strength Training: Implications for Velocity-Based Training."
Abstract
The abstract is well-structured and reflects the most relevant aspects of the study. However, including specific values could help emphasize the findings on reliability.
Introduction
The introduction is well-founded, with relevant and up-to-date references.
Methodology
- The difference in mean age between men (31.5 ± 6 years) and women (23 ± 2 years) should be mentioned, as it could affect neuromuscular response results. This should be addressed in the discussion under the limitations section.
- The participant selection criteria should be clarified, and a flowchart could be included to show the number of participants contacted and those who completed the study.
- A paragraph about the participants' prior strength training experience is recommended.
- Regarding load standardization, the study sets a minimum of ≥40 kg for men and ≥25 kg for women in the bench press, but relative intensity (% of 1RM) is not considered. Since velocity varies with intensity, shouldn't the load have been individualized based on 1RM?
- The study does not mention whether fatigue effects were controlled between trials. It should be specified whether the trial order was randomized to avoid bias due to fatigue.
Discussion
The discussion is well-structured and includes all necessary sections. However, it would be beneficial to discuss acceptable measurement errors in a training setting and provide guidance for coaches on interpreting the data, for example, when a small change in velocity indicates a real improvement.
Future Research Recommendations
It is suggested that future studies compare the LPT data with motion capture systems as a reference standard.
Finally, I believe that once these comments have been addressed, this study could be a high-quality contribution to the field of velocity-based training.
Author Response
Thank you very much for your interesting comments and contributions to improving the manuscript. The authors appreciate the value of the comments and recognize that they were important in consolidating the paper. We will also try to satisfy each of the questions.
The title is informative, but I find it too long. I recommend that the authors summarize it, for example:
"Reliability and Accuracy of Linear Position Transducers in Strength Training: Implications for Velocity-Based Training."
Response/ We would like to express our gratitude to the reviewer for this comment. We have corrected or changed to: Reliability and Accuracy of Linear Position Transducers During the Bench Press and Back Squat: Implications for Velocity-Based Training.
Abstract
The abstract is well-structured and reflects the most relevant aspects of the study. However, including specific values could help emphasize the findings on reliability.
Response/ We thank the reviewer for this comment. Yes, you are right, in abstract, we decided to avoid specificity, so as not to incur errors of repeated values, since all variables analyzed the ICCs found were higher than 0.995 with confidence intervals between 0.992-0.999. The reliability results are thus emphasized.
Methodology
- The difference in mean age between men (31.5 ± 6 years) and women (23 ± 2 years) should be mentioned, as it could affect neuromuscular response results. This should be addressed in the discussion under the limitations section.
Response/ We thank the reviewer for this comment. In the present study, the focus was directed towards the analysis of inter-device reliability rather than neuromuscular response results. With all respect, we believe that, as the devices are synchronized, the difference in mean age does not have a direct influence on the reliability results.
The participant selection criteria should be clarified, and a flowchart could be included to show the number of participants contacted and those who completed the study.
Response/ We would like to express our gratitude to the reviewer for this comment. We have added: Inclusion criteria considered: experience of more than one year of regular exercise and familiarity with the technical execution of BP and SQ. The 1 RM was also a criterion (men: BP ≥ 40 kg and SQ ≥ 60 kg; women: BP ≥ 25 kg and SQ ≥ 40 kg). These criteria were checked before and during the measurements. It was verified that no one had health problems, physical limitations, or musculoskeletal injuries that would compromise the performance of the measurements. Exclusion criteria were activated when one of these requirements was unmet, resulting in the exclusion of 6 students and the 40 subjects selected for reliability analysis.
We recognize that this type of diagram may be helpful in studies with long interventions, however, we are not sure how this may be relevant to our study, which had only a single visit for the data collection.
A paragraph about the participants' prior strength training experience is recommended.
Response/ We thank the reviewer for this comment. We have added: Inclusion criteria considered: experience of more than one year of regular exercise and familiarity with the technical execution of BP and SQ.
Regarding load standardization, the study sets a minimum of ≥40 kg for men and ≥25 kg for women in the bench press, but relative intensity (% of 1RM) is not considered. Since velocity varies with intensity, shouldn't the load have been individualized based on 1RM?
Response/ We thank the reviewer for this comment. We have explained this in the testing Procedures:
Each subject performed three attempts against previously standardized loads (men: BP = 40 kg and SQ ≥ 60 kg; women: BP ≥ 25 kg and SQ ≥ 40 kg). Standardizing the weight aims to obtain different loads for each subject, resulting in a greater trend to variability in the data.
Furthermore, considering the individualization of the load based on relative intensity (% of 1RM) could compromise the assumptions of the data's variability. This is because each percentage of 1RM has a range of associated mean propulsive velocity (1). In the present study, the load was standardized to find different intensities and, therefore, test the reliability of the devices before different velocities.
- Pareja-Blanco F, Sánchez-Medina L, Suárez-Arrones L, González-Badillo JJ. Effects of Velocity Loss During Resistance Training on Performance in Professional Soccer Players. Int J Sports Physiol Perform. 2017;12(4):512-9.
The study does not mention whether fatigue effects were controlled between trials. It should be specified whether the trial order was randomized to avoid bias due to fatigue.
Response/ We thank the reviewer for this comment. In the present study, the focus was on analyzing inter-device reliability rather than fatigue effects.
Discussion
The discussion is well-structured and includes all necessary sections. However, it would be beneficial to discuss acceptable measurement errors in a training setting and provide guidance for coaches on interpreting the data, for example, when a small change in velocity indicates a real improvement.
Response/ We thank the reviewer for this comment. Our research team will further investigate these assumptions in future studies.
Future Research Recommendations
It is suggested that future studies compare the LPT data with motion capture systems as a reference standard.
Response/ We would like to express our gratitude to the reviewer for this comment. Several investigations have explored this field, demonstrating the excellent reliability of these devices. This systematic review is exhaustive regarding these reports:
Weakley J, Morrison M, García-Ramos A, Johnston R, James L, Cole MH. The Validity and Reliability of Commercially Available Resistance Training Monitoring Devices: A Systematic Review. Sports Medicine. 2021;51(3):443-502.
Reviewer 4 Report
Comments and Suggestions for Authors
Begin with a broader context of strength training monitoring before narrowing down to velocity measurements and LPTs.
Consolidate the information on reliability metrics into a more concise paragraph, focusing on their relevance to your study.
Provide a clearer transition between the literature review and the identification of the research gap.
Explicitly state the novelty of your approach using three LPT-Cs simultaneously and how this addresses limitations in previous reliability studies.
Consider adding a brief sentence on the potential implications of your study for strength training practice and research.
It would be helpful to explain why you chose a triangulation method with three LPT-Cs over other possible alternatives.
It might add value to report additional anthropometric data, such as body fat percentage, for better context.
Consider providing more details on how the LPT-Cs and other equipment were calibrated to ensure accuracy.
It would be useful to clarify why you selected specific standardized loads and how they align with participants’ 1RM.
Justify your decision to use an effect size of 0.5 for the power analysis.
Explain the rationale behind focusing on concentric actions in your analysis.
Offer more insight into why a one-sample t-test was chosen to analyze mean differences.
Provide a more in-depth explanation of the benefits of using three LPT-Cs rather than the traditional two-device approaches. Discuss how this method enhances the robustness and validity of the reliability analysis, making your results more reliable and comprehensive.
Elaborate on the potential reasons for the small proportional bias found between the LPT-C1 and LPT-C2 interactions for ∆S. It might be helpful to consider factors such as device placement, data processing, or individual variability in subjects, and how these could contribute to the observed differences.
Help readers understand how the observed ICC, SEM, and CV values impact real-world scenarios. Discuss the practical implications for coaches, athletes, and researchers in terms of equipment selection, training protocols, and how to interpret these data in applied settings.
When comparing your findings to those of previous research, provide more detailed information about the studies you’re referencing. It would be useful to include specifics like sample characteristics, exercises performed, and the reliability metrics used, so readers can fully grasp the context of your comparisons.
Author Response
Thank you very much for your interesting comments and contributions to improving the manuscript. The authors appreciate the value of the comments and recognize that they were important in consolidating the paper. We will also try to satisfy each of the questions.
Begin with a broader context of strength training monitoring before narrowing down to velocity measurements and LPTs.
Response/ We would like to express our gratitude to the reviewer for this comment. This recommendation can be approached from a variety of perspectives. The standpoint adopted is acknowledged. However, currently, many studies address different contexts of strength training monitoring with high accuracy. To avoid the high information redundancy in the literature, we decided to be specific and limit ourselves to speed and LPT measurements.
Explicitly state the novelty of your approach using three LPT-Cs simultaneously and how this addresses limitations in previous reliability studies.
Provide a more in-depth explanation of the benefits of using three LPT-Cs rather than the traditional two-device approaches. Discuss how this method enhances the robustness and validity of the reliability analysis, making your results more reliable and comprehensive.
Response/ We would like to express our gratitude to the reviewer for this comment. The LPT-C has been previously validated when compared with T-Force (6, 10, 13) and Trio-OptiTrack (6); its reliability has also been analyzed (10, 12, 13, 23) and widely used in different investigations (25-28). However, uncertainty about the technological reliability of LPT-C remains latent. No research has yet been conducted on how the reproducibility criteria, also known as test-retest reliability and responsiveness, behave when more than two devices are considered. All the studies referenced here have used one or two versions of the same device to analyze reliability. However, none of them considered whether random error could have compromised the quality of the results. To eliminate uncertainties about the reliability of the LPT-C, the present study aimed to examine the test-retest reliability of three LPT-Cs using a simultaneous triangulation method of the same device in bench press (PB) and back squat (SQ) exercises performed on a Smith machine.
This triangulation procedure has a markedly more robust propensity to increase credibility in device reliability. This methodological procedure made it possible to compensate for the limitations previously mentioned, thus giving greater robustness to the reliability of the devices.
Consider adding a brief sentence on the potential implications of your study for strength training practice and research.
It would be helpful to explain why you chose a triangulation method with three LPT-Cs over other possible alternatives.
Response/ We would like to express our gratitude to the reviewer for this comment. The use of this triangulation procedure with the same device has a markedly more robust propensity to increase credibility in device reliability. This methodological procedure made it possible to compensate for the limitations previously mentioned, thus giving greater robustness to the reliability of the devices.
It might add value to reporting additional anthropometric data, such as body fat percentage, for better context.
Response/ We thank the reviewer for this comment. The authors preferred not to present anthropometric data because, for this type of study, these values do not provide relevant information in the reliability criteria.
Consider providing more details on how the LPT-Cs and other equipment were calibrated to ensure accuracy.
Response/ We would like to express our gratitude to the reviewer for this comment. We used only three LPT-Cs, no other devices, and maintained the original calibration criteria provided by the manufacturer.
It would be useful to clarify why you selected specific standardized loads and how they align with participants’ 1RM.
Response/ We would like to express our gratitude to the reviewer for this comment. We have explained this in the testing Procedures:
Each subject performed three attempts against previously standardized loads (men: BP = 40 kg and SQ ≥ 60 kg; women: BP ≥ 25 kg and SQ ≥ 40 kg). Standardizing the weight aims to obtain different loads for each subject, resulting in a greater tendency to variability in the data.
Furthermore, considering the individualization of the load based on relative intensity (% of 1RM) could compromise the assumptions of the data's variability. This is because each percentage of 1RM has a range of associated mean propulsive velocity (1). In the present study, the load was standardized to find different intensities and, therefore, test the reliability of the devices before different velocities.
- Pareja-Blanco F, Sánchez-Medina L, Suárez-Arrones L, González-Badillo JJ. Effects of Velocity Loss During Resistance Training on Performance in Professional Soccer Players. Int J Sports Physiol Perform. 2017;12(4):512-9.
Justify your decision to use an effect size of 0.5 for the power analysis.
Response/ We would like to express our gratitude to the reviewer for this comment. With an effect size of 0.5, an alpha value of 0.05 and a beta value of 0.90, a sample of 36 participants was required to have a 90% probability of detecting a true effect if one exists, keeping the probability of a type I error at 5%. An effect size of 0.5 is considered moderate and is large enough to be relevant in terms of a priori analysis. In addition, using an effect size of 0.5 can be consistent with previous studies, which facilitates comparison and replication of the results.
Montoro-Bombú R, Gomes BB, Santos A, Rama L. Validity and Reliability of a Load Cell Sensor-Based Device for Assessment of the Isometric Mid-Thigh Pull Test. Sensors (Basel). 2023;23(13).
Montoro-Bombu R, Field A, Santos AC, Rama L. Validity and reliability of the Output sport device for assessing drop jump performance. Front Bioeng Biotechnol. 2022;10:1015526.
Montoro-Bombú R, de la Paz Arencibia L, Buzzichelli C, Miranda-Oliveira P, Fernandes O, Santos A, et al. The Validity of the Push Band 2.0 on the Reactive Strength Index Assessment in Drop Jump. Sensors. 2022;22(13):4724.
Explain the rationale behind focusing on concentric actions in your analysis.
Response/ We would like to express our gratitude to the reviewer for this comment. We have explained in the introduction: “Research (Sanchez-Medina L, Gonzalez-Badillo JJ ) has focused more on the velocity of the concentric phase in an exercise, and it has been suggested as a valid approach for measuring and adjusting these performance parameters”. Because concentric actions are the most analyzed in the literature on athletic performance, our research focused only on the analysis of concentric actions. Other studies may be aimed at analyzing eccentric actions.
Sanchez-Medina L, Gonzalez-Badillo JJ. Velocity loss as an indicator of neuromuscular fatigue during resistance training. Med Sci Sports Exerc. 2011;43(9):1725-34.
Elaborate on the potential reasons for the small proportional bias found between the LPT-C1 and LPT-C2 interactions for ∆S. It might be helpful to consider factors such as device placement, data processing, or individual variability in subjects, and how these could contribute to the observed differences.
Response/ We wish to express our gratitude to the reviewer for this comment. We believe this statistical bias may be tied to small errors inherent to the device. In our discussion, we address this issue succinctly, as we do not consider it a significant problem in a real sports context. I quote, “For PB MD (0.31 ± 0.47; CI 0.24–0.37), it barely represents, in practical terms, a difference of 3 millimeters, which fully corresponds with the results reported in graph 1. We recommend that practitioners utilize this SEM data as the range within which a displacement error (0.3 millimeters) might be expected, which in practical training terms may not be significant for coaches but could be for a sports scientist considering this measure as a meaningful difference for their research. We also recommend using the SDC criteria reported here to identify the minimum change in ∆S within a series (BP = 0.10 cm; BQ = 0.13 cm). "
Help readers understand how the observed ICC, SEM, and CV values impact real-world scenarios. Discuss the practical implications for coaches, athletes, and researchers in terms of equipment selection, training protocols, and how to interpret these data in applied settings.
Response: We would like to express our gratitude to the reviewer for this comment. After analyzing our study, we believe that paragraphs 2 and 3 of our introductions provide a direct answer to the reviewer's question. In our opinion, we justify why the ICC, SEM, and CV are the most used values in the literature.
When comparing your findings to those of previous research, provide more detailed information about the studies you’re referencing. It would be useful to include specifics like sample characteristics, exercises performed, and the reliability metrics used, so readers can fully grasp the context of your comparisons.
We would like to express our gratitude to the reviewer for this comment. The authors acknowledge the importance of this commentary in the context of other observational or experimental comparison studies. However, we question whether including the details requested would add scientific value to the discussion of our results. If the criteria for data variability are maintained in any study, the reliability and reproducibility of a device are independent of the sample characteristics and the exercises performed. When two devices are properly synchronized, they should provide consistent measurements regardless of individual subject variability, environmental factors, and so on. The reliability metrics reported by other studies (ICC, SEM, and CV) were comparable to ours, as they followed the same statistical procedures we employed.
Round 2
Reviewer 3 Report
Comments and Suggestions for Authors
Thank you very much for your thorough review and for taking into account all the proposed comments. With these corrections, the article has improved in quality, so I believe it can be published in its current format.
Reviewer 4 Report
Comments and Suggestions for Authors
All comments have been addressed by the authors.